# Effectual Environmental Enrichments for Commercial Broiler Chickens

**DOI:** 10.3390/ani15192829

**Published:** 2025-09-28

**Authors:** Seong W. Kang

**Affiliations:** Department of Poultry Science, Center of Excellence for Poultry Science, University of Arkansas, Fayetteville, AR 72701, USA; swkang@uark.edu

**Keywords:** enrichment hut, variable light intensity, hepatic fatty acid, welfare, mental health

## Abstract

**Simple Summary:**

Voluntary movement may improve animal welfare and promote sustainable growth. After providing lower-light-intensity areas with dual lighting intensities and enrichment huts in commercial broiler houses, broiler chickens exhibited better welfare indicators in their legs, livers, and eyes, suggesting that midday resting in these areas improved the welfare and health of the animals. In addition, the mental status of lighting-enrichment-treated birds indicated lower stress susceptibility compared to birds reared in conventional lighting program houses. The growth performance, measured by the daily weight gain and feed conversion ratio, was significantly better than with traditional lighting programs.

**Abstract:**

Environmental enrichment, such as lighting, has affected the behaviors, welfare, and production of commercial broiler chickens. However, most studies have focused on constant light intensities to determine their effect on welfare and performance. Research indicates that the significant contrast of light intensities in broiler houses promotes pronounced daily patterns of behavior and activity, impacting broiler chicken health. Birds exhibited preference behaviors in bright-intensity light during active behaviors, such as eating and drinking, but in darker areas when resting. Light intensity preferences may be associated with the voluntary instinctive movement of birds by providing choices for birds. Increasing broiler chickens’ movement may boost welfare, especially leg health, which is a leading cause of culling and late mortality in commercial production. In this review, we discuss the progress and results of practical environmental enrichments, enrichment lighting, and huts in commercial broiler houses. We briefly address interpretations of improved welfare and performance and suggest directions for future research that may interest poultry scientists.

## 1. Introduction

Diurnal animals rest at night and are active during the day to engage in feeding, playing, mating, and other activities. The disruption of sleep/resting at night or when animals need sleep/resting during the daytime causes oxidative stress and abnormal physiological responses [1,2]. Sleep is a time when the brain starts a cleaning process to flush out waste and toxins it accumulates during wakefulness [3]. It is well known that sleep or resting can replenish antioxidants in the body and improve the oxidative stress status of animals [4]. In humans, Wiesner et al. (2018) investigated the effects of a brief period of daytime sleep (nap) and endogenous melatonin on reward learning [5], showing that exogenous melatonin modulates the responses of the dopaminergic reward system and acts as a neuroprotectant promoting memory. An interesting finding of their study is that a daytime nap in darkness increases the saliva melatonin level [5]. In addition, a midday nap impacts physical performances in humans [6], suggesting that the midday nap is a part of the endogenous biological rhythms, a compensatory response to sleep loss, or a behavioral response to exogenous factors.

Light is an influential environmental factor inducing diverse physiological functions, including brain functions [7]. In commercial broiler chickens, light is an environmental factor that influences behavior, welfare, and productivity [8,9,10,11]. Light intensity has been reported as one of the most noticeable effects of light on birds’ behavior [9,12,13]. Light intensity encourages broiler chickens’ movement, though research examines constant light levels and their impact on welfare and performance. More interestingly, Blatchford et al. (2012) reported a strong effect of light intensity contrast on the behavior and health of broiler chickens [12], suggesting that a higher contrast in light intensity is associated with significant daily rhythms of behavior. Multiple studies indicated that broiler chickens demonstrate increased movement and activity at higher light intensities [14,15,16].

Accordingly, in commercial broiler production houses, providing specific environments where birds can comfortably consume their feed and water, play, rest, and nap/sleep as they choose has been very appealing and interesting for their welfare. Here, we provide a summary of recent research findings regarding the effects of variable lighting programs on animals and review the progress of environmental enrichment, lighting, and hut programs in commercial broiler houses, and we introduce the basic research results and conceivable explanations regarding the physiological mechanisms underlying the effects of the variable lighting and hut programs on the welfare and performance of broiler chickens.

## 2. Behavioral Preference of Broiler Chickens for Variable Light Intensity

It is generally presumed that we are aware of our movements mainly because we can sense ourselves moving as ongoing peripheral information coming from our muscles and retina reaches the brain [17,18]. Recent evidence, however, suggests that, contrary to common views, conscious intention to move is independent of movement execution per se [18]. Studies show that dopamine shapes neuronal responses and modulates aspects of attentional processing in a region in the brain involved in spatial awareness and orientation, receiving inputs from various brain areas [19].

Chickens are naturally active animals that show natural behaviors, including dustbathing, scratching, and pecking. Sedentary behaviors, characterized by a lack of physical activity, are often attributed to the limited space or confinement in commercial broiler houses. Lighting has been suggested to be an attractive way to improve the locomotor activity of broiler chickens [11]. In addition to supporting vision, light has wide-ranging effects on the biological functions of animals, including behavioral and physiological processes. Studies in mammals show that a complex visual system is required to adjust animals to the light environment [20,21]. In addition, researchers have demonstrated that some regions of the brain are involved in cognitive processing and mood, indicating the sensitivity to light intensity of the photosensitive retinal ganglion cells [22,23].

Light is an influential environmental factor influencing brain functions and consequently affecting behavior; however, the precise circuits that mediate the effects of light on behaviors remain unclear [7]. By receiving environmental lighting inputs, animals integrate information associated with the external environment, internal brain states, and action choices, allowing animals to induce appropriate behaviors depending on their age, sex, endogenous hormonal status, and memories [24]. Light/dark light preference behavior is a highly conserved innate behavior throughout the animal kingdom and is vital for animal survival [25,26,27,28]. Two pairs of neurons in the central brain and neuropeptide-releasing neurons were reported to control dark preference behavior in invertebrates [26,28]. Exposure to an environment with favorable illuminance may help the animal to forage for food, avoid predators, find conspecifics, and perform courtship behaviors [29].

Although light/dark preference paradigms have been widely used to evaluate behaviors of animals [30], the neural circuit underlying light/dark preference in vertebrates is mainly unknown, except that the involvement of the retina and the minor contribution of deep-brain photoreceptors being reported [31]. The ventral tegmental area (VTA) of the midbrain is associated with positive affective states [24]. A light-responsive neural circuit that relays the environmental light signals to regulate feeding behavior and innervates dopamine neurons in the VTA was reported recently in rodents [32,33]. The reward and food-seeking system consists mainly of dopaminergic neurons in the VTA of the midbrain that may be applicable in birds, given that deep-brain photoreceptor melanopsin was identified in the VTA of the avian brain [34,35].

When broiler chickens had choices of different light intensity areas in the preference studies, birds showed specific preference behaviors [14,36]. Studies have shown that broiler chickens generally prefer brighter light. When given a choice, they tend to choose higher-intensity light during active behaviors and dimmer areas when resting. The variation of locomotor activity during an alternating light schedule is higher than in constant light, which may suggest more synchronization of behavior in the group. In addition, low light intensity results in a significant reduction of movement and an increase in resting time. Consequently, low light intensity can negatively affect leg health, produce hock burn, and cause breast blisters. On that account, it has been strongly suggested that increasing the locomotor activity of broiler chickens could improve their welfare in commercial broiler houses [36,37,38].

It is crucial to find ways of providing animals with what they value for their biological necessity [39]. Commercial broiler chickens have been raised in relatively low-light-intensity houses with low locomotor activity, with limited stimulation and consequent issues such as leg health (Figure 1A). Bright light has been suggested to improve the welfare of birds, because broiler chickens show more pronounced daily rhythmic behavior and appear to display more comfort behaviors under brighter light [8]. Overall, birds show more active and energetic behavior when reared with high-intensity compared to low-intensity light [40,41]. Light intensity has been shown to affect the activity of birds, but studies mainly focused on constant light intensities to establish their effect on welfare. In the preference study in the pen of a commercial broiler chicken farm, birds showed preference for the higher-light-intensity area when they were performing active behaviors but preferred dimmer areas when resting (Figure 1B) [15,42], suggesting that the behavioral stimulation by different light intensities may generate the motivation for better movement by their light preference through receiving contextual visual information.

## 3. Effect of Daytime Resting in Darkness on the Welfare of Broiler Chickens

While it has long been recognized that light exposure suppresses melatonin production from the pineal gland during the night, there was little current evidence that exposure to darkness and sleep during the daytime could increase melatonin levels [45]. Studies have shown that melatonin is significantly better than the classic antioxidants in resisting free-radical-based molecular destruction [45,46,47]. A low level of melatonin is associated with an increased risk of diseases and neurodegenerative disorders [48,49,50]. Exposure to darkness during the daytime has not generally been reported to increase circulating melatonin concentrations [51]. Interestingly, the effect of daytime napping on the mood and antioxidant defense was reported in a human study, suggesting that napping in darkness was associated with higher antioxidant defense and improvement in physical and cognitive performance [52,53]. Studies have reported that unconventional melatonin profiles in the circadian afternoon have been observed in healthy participants who sleep during the day under highly controlled laboratory conditions or take a short nap [5,54].

Melatonin performs various antioxidant functions, such as being a scavenger of free radicals. It possesses a remarkable ability to rescue dopaminergic neurons from cell death in several experimental models of oxidative stress [55,56]. Exogenous melatonin modulates the responses of the dopaminergic reward system in the VTA and acts as a neuroprotectant [5]. Melatonin receptors were present in the dopaminergic neurons of the VTA, and melatonin induced tyrosine hydroxylase expression in VTA dopaminergic neurons, indicating that melatonin influences neurotransmitter systems involved in addiction, such as the dopaminergic, glutamatergic, serotonergic, and endogenous opioid systems in the VTA [57,58,59].

Commercial broiler houses were tested for preference behavior, physiological changes, and performance by providing variable-lighting (VL) areas to birds (Figure 1C). Broiler chickens in the VL house had an average 2.2% lower feed conversion ratio compared to those in the 20 lx house (Table 1) [60]. In VL broiler houses, active resting and napping behaviors of birds were observed (Figure 1C), which suggests a daytime nap in darkness may elicit a melatonin response and that the magnitude of this response positively correlates with the subsequent physiological responses of broiler chickens observed in the commercial houses [35].

To identify the most advantageous environmental enrichments (EEs) for commercial broiler chickens, three commercially available EEs (board, hut, and ramp) were tested, and environmental enrichment huts (EHs) proved to be the most favorable enrichment for broiler chickens (Cobb 700, mixed sex) with three huts/92.9 m^2^ (1000 sq ft) [61]. Therefore, EHs were tested to provide an additional way to provide low-light-intensity areas for birds (Figure 1D), showing the synergistic effect on ocular welfare (Table 2). A plausible mechanism for this may be an effect of resting and sleeping on daytime melatonin production by using additional enrichment huts [45].

## 4. Improved Welfare and Performance by Environmental Enrichment Lighting and Huts in Commercial Broiler Houses

Sedentary behavior and physical inactivity during the daytime are associated with weight gain, muscle weakness, leg health issues, and elevated risk of developing metabolic syndrome in animals, including humans [62,63,64,65]. Active walking and physical exercise are evidence-based treatment strategies to improve broiler chickens’ leg health and a universally accepted measure of the welfare and health of animals [36,37,39,66]. Active movements would be a voluntary choice to satisfy animals’ innate biological needs and playfulness. Voluntary choice is the decision-making process where animals act based on their preferences and perceived benefits rather than being compelled by external compulsion [67]. Therefore, it is vital to promote voluntary activities for animals in commercial broiler houses.

Leg health is one of the most prevalent reasons for culling and mortality during the grow-out of commercial broiler chickens. There is increasing evidence to suggest that increasing locomotor activity in broiler chickens may improve their leg welfare [36,37,42,66]. Previous VL studies (Figure 1B) indicate that when birds have a dual light choice, they consume more feed in the brighter area compared to the darker area [15,42]. A recent study reported that providing different lighting intensity areas in commercial broiler houses stimulates voluntary walking behavior for consuming feed/water and taking rest, as well as improving leg health and performance (Figure 1C) [60]. Interestingly, dopaminergic systems in the brain are involved in the regulation of voluntary physical activity and behavior [68,69], indicating that VTA dopaminergic neurons in the midbrain are likely associated with voluntary physical activity and behavior of animals [70,71]. A predictive report indicates that non-retinal light perception in the dopaminergic system (VTA) of the midbrain in the poultry brain [35].

It is known that during physical activity, the skeletal muscles of the legs release myokines such as brain-derived neurotrophic factor (BDNF) and interleukin-6 (IL-6), which not only support muscle recovery and cognitive function but also maintain the integrity and functional stability of bones as endocrine organs [72,73,74]. In addition, physical exercise leads to an increase in antioxidant protection in animals by upregulation of antioxidant enzyme activity in an exercise-type-dependent manner [75,76]. When commercial broiler chickens were raised with light intensity choices between bright and dim areas, a lower number of culled birds with leg problems was observed compared to 20 lx constant light intensity houses [60], suggesting that augmented natural movement behaviors appear to improve leg health and cause a consequent reduction of mortality in broiler chickens. In addition, it was observed that the improvement of the daily weight gain (DWG) and the feed conversion ratio (FCR) was observed in the broiler chickens raised in VL houses (Table 1), indicating the significant economic benefit of the VL program in broiler chicken production. One of the possible explanations for the beneficial effects of voluntary activity and exercise on leg health may be the role of the exercise-induced output and secretion of myokines [77,78].

Physical exercise has a protective effect on metabolic-dysfunction-associated fatty liver disease [79,80,81]. Emerging evidence accentuates the revolutionary role of physical exercise as a potent nonpharmacological intervention for durable metabolic improvements [81]. The liver is the central metabolic organ responsible for maintaining blood glucose levels and controlling metabolic homeostasis, acting as the primary site for lipid metabolism. Hepatic lipid metabolism is closely associated with glucose metabolism in liver diseases [82,83]. The liver stores glucose in the form of glycogen and releases glucose into circulation by either glycogenolysis or gluconeogenesis [84,85,86]. In addition, the liver is the principal detoxifying organ and maintains metabolic homeostasis, but sedentary behavior causes hepatic metabolic dysregulation in animals [87]. Under normal conditions, the liver regulates lipid metabolism to meet systemic energy needs by the tightly regulated processes of fatty acid uptake, synthesis, and oxidation pathways [88,89]. If these processes become dysregulated, excess lipid accumulation can occur [90]. Sedentary behavior leads to hepatic lipid accumulation, stimulates de novo lipogenesis, and facilitates hyperlipidemia [90].

Recent VL and EH studies indicate the improved metabolic syndrome in the liver of broiler chickens by environmental VL and EH programs [61], suggesting that EHs significantly stimulated fatty acid β-oxidation in the liver of VL and EH house birds compared to 20 lx house birds, and that the VL program decreased fatty acid synthesis (downregulation of acetyl-CoA carboxylase alpha (ACCα)) and activated fatty acid β-oxidation (upregulation of carnitine palmitoyl-transferase 1a (CPT1a)) in the liver of broiler chickens (Figure 2). These results suggest that the stimulated voluntary activity by the VL program improved hepatic lipid metabolic function and may indicate efficient nutritional support and use for the growth of broiler chickens.

It is reported that low light intensities have been found to cause abnormal behaviors, poor eye development, and dysregulated growth of animals, including broiler chickens [9,91,92,93,94]. The vertebrate retina produces melatonin in a light-dependent rhythmic fashion, but independent of the rhythm of the production of the hormone in the pineal gland, although to a much lesser extent than in the pineal gland [95,96,97]. When animals have a short period of daytime sleep or nap in darkness, it may increase melatonin production, with a modest melatonin elevation [5,51,54]. These unconventional melatonin profiles, unlike a nocturnal unimodal profile, have been reported to show elevated melatonin levels in the circadian afternoon [5,54].

Melatonin produced during the daytime may act directly on ocular structures to mediate a variety of diurnal rhythms and physiological processes within the eye, including the modulation of intraocular pressure [98]. Melatonin synthesized in the retina comes exclusively from the photoreceptor cells [99,100,101]. These photosensitive retinal ganglion cells are responsible for mediating the light/dark cycles and regulating melatonin’s production and secretion in the retina [97,102]. Several studies have shown that melatonin displays strong antioxidative properties, which play a protective role in many pathologies associated with oxidative stress. Oxidative stress induced by the stress status of animals causes ocular dysfunctions, including structural deformation and the dysfunctional development of the eyes.

Retinal oxidative stress induces high intraocular pressure (IOP), which is a significant risk factor for glaucoma and a major cause of vision loss [103,104]. Glaucoma can be induced in an animal model, domestic chicks, by rearing them under continuous light, which causes increased eye weight, globe enlargement, decreased corneal diameter, and all effects associated with light-induced avian glaucoma. Additionally, exposure to very dim light causes eye enlargement in avian species [105,106]. A significant decrease in retinal antioxidant defense was observed in an animal model mimicking human open-angle glaucoma, and melatonin ameliorated glaucoma [107]. This is also justified by a hypotensive effect of melatonin resulting in a reduction of the IOP in a glaucoma avian model [108].

In broiler chickens, the corneal diameter (CD) and eye weight (EW) are inversely correlated with the ocular health of broiler chickens [9,94]. The average CD of the adult chickens (White Leghorn) is 9.1 ± 0.1 mm horizontally, and changes with age [108]. The CD was significantly greater in the low-light-intensity-treated broiler chickens than in the brighter-light-intensity-treated birds with heavier eyes [9,92]. EHs reduced CDs in 20 lx and VL birds (Table 2), suggesting an improvement in eye health and indicating the beneficial effect of EHs on eye health in the VL birds. It appears that when birds are given the lower-light-intensity areas by VL and EHs during the daytime, it may help the normal eye development compared to the 20 lx house by providing non-circadian light/dark adaptation time.

## 5. Future Perspectives

Augmented voluntary behaviors by VL and EHs appear to improve leg health and cause a consequent reduction of mortality in broiler chickens, as observed in the improvement of the DWG and FCR. One possible explanation for the beneficial effects of voluntary activity and exercise on leg health is the role of exercise-induced output and secretion of myokines, which has not been studied in poultry research. A recent study using different light intensities in barns was performed to investigate the effects of various light intensities on welfare [109], indicating that birds need enough distance and space for their movement, like commercial chicken houses, to achieve adequate performance.

Exercise-induced myokines may exert preventative or curative effects against muscle wasting, while intrahepatic and circulating triglycerides appear to be key factors in chicken breast myopathy [110,111,112]. A moderate increase in CPT1a activity can significantly lower liver triglyceride levels by boosting fatty acid oxidation [113]. Therefore, stimulation of fatty acid β-oxidation may be satisfactory to reduce hepatic triglyceride, which is one of the critical biochemical components involved in breast myopathy in broiler chickens [113]. Further research is required to investigate the possible beneficial effects of environmental EL and EHs on breast myopathy in commercial broiler chickens.

## 6. Conclusions

Effectual environmental enrichments, VL and EHs, for commercial broiler chickens were established and verified by investigating their preference behaviors, welfare indicators, physiological responses in the liver and brain, and growth performance. A brief period of daytime resting in the darker areas provided by VL and EHs appears to stimulate birds’ voluntary movement and improve physiological and mental benefits, as well as the welfare of birds. Increasing broiler chickens’ movement through these environmental enrichments enhances performance and economic benefits in commercial broiler chicken production.

## Figures and Tables

**Figure 1 animals-15-02829-f001:**
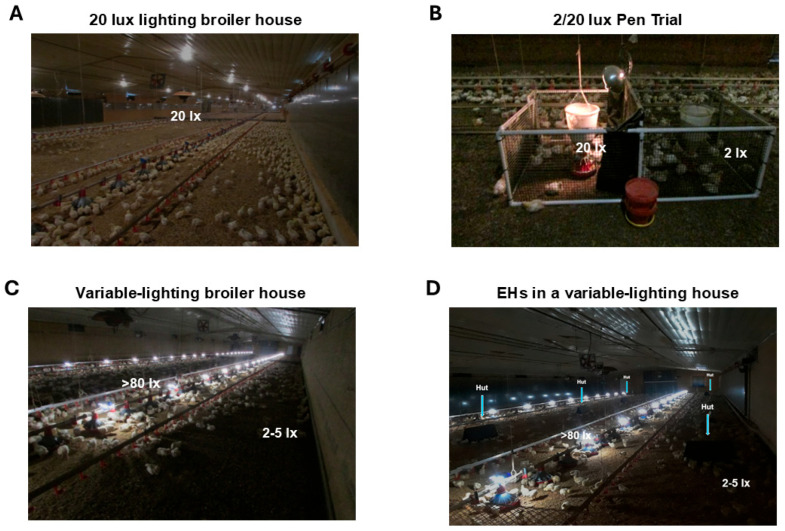
Investigation of the effects of environmental enrichment, lighting, and huts in commercial broiler houses [43,44]. (**A**) Conventional continuous 20 lux (lx, measured on the birds’ heads) of light intensity in the broiler chicken house (Tyson Foods Broiler Welfare Research Farm, Hindsville, AR, USA). (**B**) Preference testing pen (121.9 × 121.9 cm, Applied Broiler Research Farm, University of Arkansas, Savoy, AR, USA) was constructed from a PVC frame and plastic-coated wire, the walls of the pen were covered in black plastic film to exclude light, and the pen was divided into two identical compartments using a divider that was raised to bird height to allow bird passage under the divider yet prevent light penetration from the hanging light above (2/20 lx). (**C**) Average light intensity around feed lines was over 80 lx, and the light intensity in the middle and sidewalls of the house was about 2–5 lx. (**D**) Enrichment huts (EHs; 35.6 cm (W) × 76.2 cm (L) × 35.6 cm (H)) were placed in the variable-lighting houses (3 EHs/92.9 m^2^ (1000 ft^2^)).

**Figure 2 animals-15-02829-f002:**
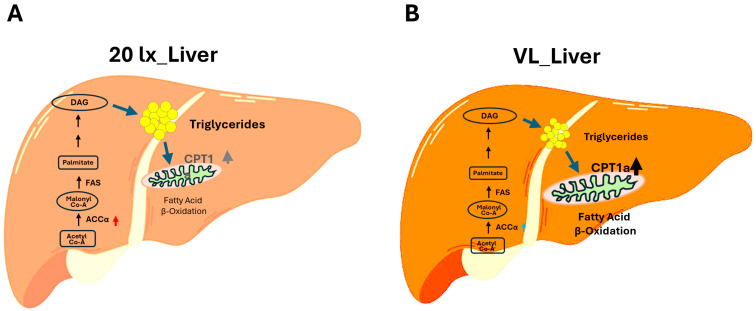
Increase in fatty acid β-oxidation and decrease in fatty acid synthesis in the broiler chickens’ liver affected by a VL program. (**A**) Liver of birds raised in the conventional continuous 20 lx of light intensity in the broiler chicken house. (**B**) Liver of birds raised in the VL program (average light intensity around feed lines was over 80 lx, and light intensity in the middle and sidewalls of the house was about 2–5 lx). Expression of ACCα in the liver was downregulated in VL birds compared to 20 lx birds. Expression of CPT1a was significantly increased in the liver of broiler chickens raised in the VL houses. ACCα: Acetyl-CoA carboxylase alpha catalyzes the carboxylation of acetyl-CoA to malonyl-CoA, the first step in de novo fatty acid biosynthesis in the liver. FAS: Fatty acid synthase catalyzes the de novo synthesis of fatty acids. DAG: Diacylglycerol, an increase in hepatic DAG content results from an imbalance in rates of fatty acid uptake relative to rates of mitochondrial fatty acid oxidation and conversion of DAGs to triglycerides. CPT1a: Carnitine palmitoyl-transferase1a is the rate-limiting enzyme in fatty acid β-oxidation, and its deficiency or abnormal regulation can result in metabolic disorders.

**Table 1 animals-15-02829-t001:** Effects of different lighting programs on the production performance of commercial broiler chickens.

Trial	Treatment	DWG (g)	FCR
Trial 1 (56 days)	5 lx	67.1	1.96
	20 lx	66.3	1.98
	NL	65.7	2.00
	VL	67.3	1.93
Trial 2 (51 days)	5 lx	66.4	1.86
	20 lx	64.2	1.84
	NL	64.0	1.87
	VL	66.5	1.82
Trial 3 (49 days)	5 lx	63.9	1.92
	20 lx	66.2	1.91
	NL	61.6	1.99
	VL	66.3	1.89
Trial 4 (55 days)	20 lx	69.0	1.97
	20 lx	67.6	2.07
	VL	68.7	1.94
	VL	70.6	1.95

Daily body weight gain (DWG) and feed conversion ratio (FCR) were obtained from the processing plant at the end of each trial for the whole house. The average daily weight gain of the birds in the VL house was 4.3, 1.2, and 4.1% higher than that in 5 lx, 20 lx, and natural light (NL) houses, respectively. From the feed conversion ratio (FCR) of four trials, the average FCR of 20 lx-treated and VL-treated birds was 1.95 and 1.91, respectively. The average FCR of the VL-treated birds was 2.2% lower than that of 20 lx-treated birds [36,44]. Partially adapted from Table 2 in [60].

**Table 2 animals-15-02829-t002:** Effects of environmental enrichments (VL and EHs) on the broiler eye.

Treatment	BF (mm)	SS (mm)	CD (mm)	CD/BW (mm/0.1 kg)	EW (g)	EW/BW (g/kg)
20 lx_Con	16.107 ^a^	19.246 ^a,b^	9.343 ^a^	1.056 ^a^	2.824 ^a^	0.319 ^b^
20 lx_Hut	15.784 ^b^	18.934 ^b^	9.039 ^b^	1.088 ^a^	2.641 ^b^	0.318 ^b^
VL_Con	15.852 ^a,b^	19.352 ^a^	9.194 ^a^	1.055 ^a^	2.916 ^a^	0.335 ^a^
VL_Hut	15.797 ^a,b^	19.357 ^a^	8.818 ^b^	0.996 ^b^	2.774 ^a,b^	0.313 ^b^

The eye weight was obtained from a sample of birds in trial 4. Twelve birds per lighting/hut treatment houses (20 lx_Con, 20 lx_Hut, VL_Con, and VL_Hut) were weighed and then killed via cervical dislocation. The eyes were removed, and the adhering tissue was trimmed from each eye. The anteroposterior size (back to front, BF), mediolateral diameter (side to side, SS), corneal diameter (horizontal, CD), and weight of each eye were measured using a digital caliper and a digital scale. CD/BW: corneal diameter was adjusted for body weight (mm/0.1 kg). EW/BW: eye weight was adjusted for body weight (g/kg). ^a,b^ Means within a column with different superscripts differ significantly (*p* ≤ 0.05) [36].

## Data Availability

The original contributions presented in this study are included in the article material. Further inquiries can be directed to the corresponding author.

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
