# Peer review of "Effectual Environmental Enrichments for Commercial Broiler Chickens"

_animals, 2025, doi:10.3390/ani15192829_

Round 1

Reviewer 1 Report

Comments and Suggestions for Authors

According to the latest report by the Organisation for Economic Co-operation and Development (OECD) and the Food and Agriculture Organisation of the United Nations (FAO), global meat production is likely to increase by 13% to an estimated 406 million tonnes per year by 2034. At the same time, poultry consumption is also expected to increase by 21%. Poultry is the fastest growing segment of the meat market, and forecasts indicate that by 2029 it will account for 62% of additional global consumption. At the same time, poultry production requires fewer resources and has less impact on the environment, which is in line with the growing number of environmentally conscious consumers.  

Broiler chicken production requires specific technical conditions to be met, including bird density , adequate temperature, constant access to fresh water and feed, a ventilation system to ensure air quality (control of ammonia, hydrogen sulphide, CO2), but also adequate lighting, dry and clean litter, hygiene and regular disinfection of the premises. Improving conditions, e.g. through appropriate lighting, affects the behaviour, welfare and production of broiler chickens in commercial farming. Increased consumption and consumer awareness require improvements in the production process. A synthetic discussion of the importance of one of the key production parameters provides an interesting compendium of knowledge.

Suggestions

No separate section on Materials and methods, criteria for selecting manuscript cited in the article.

Suggestions

Abstract

Line 20 - Can lighting in a henhouse be considered environmental enrichment? Please explain.

Introduction

Line 69 - Please correct ‘enriched lighting’ to ‘variable lighting’. Enriched environments usually consist of additional elements that attract attention and enable activity and play.

No separate section on Materials and methods, criteria for selecting studies cited in the article.

Please indicate the databases to be searched, keywords, and criteria for verifying articles.

Please provide the source material:

Figure 1 Line -111-114;

Table 2 line - 272

Please supplement your work with more examples of elements that enrich the building environment.

The conclusions require correction; their content is not based on the article.

Author Response

Dear Reviewer:

We appreciate your helpful and detailed feedback and comments, and hope that the explanation below has thoroughly addressed all your comments. In this response, we discuss each of your comments individually, along with corresponding responses.

In the revised manuscript, we used the latest version of the manuscript (manuscript v3.docx).  Two tables were converted into editable MS Word tables, rather than images. Two additional references were added, and one reference was changed in position. All references were confirmed. Yellow-marked words or paragraphs were the ones that were changed in the revised manuscript.

# Reviewer 1

Comment 1: No separate section on Materials and methods, criteria for selecting manuscripts cited in the article.

Response 1: This manuscript aims to provide a summary of the variable-lighting and enrichment huts programs.  We focused on providing information on specific environmental enrichments that applied to commercial broiler production. In the present manuscript, the authors describe methodologies with references, and there are details in the legends of each figure and table. Figure 1 legend lines 111-121 with two additional references [37,38]; Figure 2 legend lines 275-287; Table 1 legend lines 214-219 with three references; Table 2 legend lines 311-318 with one reference in the revised manuscript.

Comment 2: Line 20 - Can lighting in a henhouse be considered environmental enrichment? Please explain.

Response 2: I got the same question at several symposia. Yes, I think lighting in a henhouse can be considered environmental enrichment. However, it should be optimized appropriately, not just by using different lighting intensities. The main point of the variable-lighting program is to provide comfortable areas in the big commercial broiler houses for the real-time needs of birds. When birds need to eat and drink, they need bright light stimulation for their comfort, and when they need to rest, the darker areas are preferable and comfortable, as discussed in the manuscript. Unfortunately, we lack supporting evidence for this hypothesis in avian studies; however, studies in mammals cited in this manuscript can provide valuable insights.

Comment 3: Line 69 - Please correct ‘enriched lighting’ to ‘variable lighting’. Enriched environments usually consist of additional elements that attract attention and enable activity and play.

Response 3: Thank you for bringing this up. As suggested, “enriched lighting” was changed to “variable lighting” through the revised manuscript.  Also, EL was changed to VL in the revised manuscript.
Accordingly, “Enriched light” in Keywords was deleted.  ELs in Figures 1, 2, Tables 1, and 2 were changed to VL.  All changes were yellow-marked in the revised manuscript.

Comment 4: Please indicate the databases to be searched, keywords, and criteria for verifying articles.

Response 4: In this manuscript, the authors did not systematically search the databases and keywords. The references cited were from the individual section theme that we addressed and the authors’ previous research. As you can see in the manuscript, many references come from mammalian studies due to the scarcity of poultry studies.

Comment 5: Please provide the source material: Figure 1 Line -111-114; Table 2 line - 272.]

Response 5: Agree. In the revised manuscript, the source materials and references were presented in lines 111-121 for Figure 1 and line 318 for Table 2.

Comment 6: Please supplement your work with more examples of elements that enrich the building environment.

Response 6: Thank you for this comment. We tested three commercially available environmental enrichments (board, hut, and ramp) in the commercial broiler houses and achieved better results in the EH houses. Then, we tried different numbers of huts and found that three huts / 1000 sqft were optimum compared to 1 and 6 huts (Kang et al., 2024). In the revised manuscript, this information was added in lines 179-182.

Comment 7: The conclusions require correction; their content is not based on the article.

Response 7: Thank you so much for pointing this out. We have, accordingly, revised the manuscript. The Future Perspectives (lines 339-355) and Conclusions (lines 356-364) sections were divided and revised to enhance clarity for readers in the revised manuscript.

Reviewer 2 Report

Comments and Suggestions for Authors

The overall topic of this literature review is very interesting and, given the growing interest in the welfare of broiler chickens, it is an important issue. The review is written to a high standard and demonstrates the authors' expertise in this field. However, to improve the quality of this work, I would recommend a few points which, in my opinion, would increase its practical impact. 

I would add a section focusing on studies that examined different lengths of daylight hours, with an emphasis on physiological parameters and performance parameters in broiler chickens. These studies could also be summarised in a table.

The main problem, however, is the focus of the article, as it is presented as a review, but there are passages presenting the results of studies.

The paper presents some study results, but essentially without methodology. If certain results are mentioned in the paper, the methodology should also be mentioned. The tables show significant differences between the groups, but there is no mention of statistical processing anywhere. I therefore recommend either deleting these practical results from the paper altogether, as it is a review, or at least briefly describing the methodology used to conduct the experiment.

I recommend deleting Figure 2, as it shows something that does not need to be shown in the figure.

The conclusion summarises the results of the study and cites articles. The conclusion of a review should summarise the authors' opinion on the issue, and citations of articles are not necessary here. The conclusion should therefore be rewritten.

Author Response

Dear Reviewer:

We appreciate your helpful and detailed feedback and comments, and hope that the explanation below has thoroughly addressed all your comments. In this response, we discuss each of your comments individually, along with corresponding responses.

In the revised manuscript, we used the latest version of the manuscript (manuscript v3.docx).  Two tables were converted into editable MS Word tables, rather than images. Two additional references were added, and one reference was changed in position. All references were confirmed. Yellow-marked words or paragraphs were the ones that were changed in the revised manuscript.

# Reviewer 2

Comment 1: I would add a section focusing on studies that examined different lengths of daylight hours, with an emphasis on physiological parameters and performance parameters in broiler chickens. These studies could also be summarised in a table. The main problem, however, is the focus of the article, as it is presented as a review, but there are passages presenting the results of studies.

Response 1: The authors appreciate your pointing this out. In this article, the authors focused on providing information on specific environmental enrichments that applied to commercial broiler production rather than summarizing the information from lighting studies.  The aims of this manuscript were clearly described in lines 64-70 of the revised manuscript.

Comment 2: The paper presents some study results, but essentially without methodology. If certain results are mentioned in the paper, the methodology should also be mentioned. The tables show significant differences between the groups, but there is no mention of statistical processing anywhere. I therefore recommend either deleting these practical results from the paper altogether, as it is a review, or at least briefly describing the methodology used to conduct the experiment.

Response 2: In the present manuscript, the authors summarize the recently published articles and conferences on the variable lighting and enrichment hut studies. In terms of methodology, the authors describe the details in the legends of each figure and table. Figure 1 legend lines 112-122 with two additional references [37,38]; Figure 2 legend lines 276-288; Table 1 legend lines 215-220 with three references; Table 2 legend lines 312-319 with one reference in the revised manuscript.

Comment 3: I recommend deleting Figure 2, as it shows something that does not need to be shown in the figure.

Response 3: Figure 2 is critical visual information and the main summary to show the effects of the variable-lighting program on the physiological adaptation that occurred in the liver. The authors would like to present this figure to show the beneficial effect of voluntary exercise stimulated by a variable-lighting program in the commercial broiler house, indicating the upregulation of fatty acid beta-oxidation, one of the critical metabolic pathways in broiler chickens. In the revised manuscript, this figure is well described in lines 266-275 and should be presented to readers. Figure 2 has improved in quality for readers between lines 275 and 276.

Comment 4: The conclusion summarises the results of the study and cites articles. The conclusion of a review should summarise the authors' opinion on the issue, and citations of articles are not necessary here. The conclusion should therefore be rewritten.

Response 4: Thank you for this valuable comment and for drawing this matter to our attention. We agree with this comment. Therefore, in the revised manuscript, the Future Perspectives (lines 340-356) and Conclusions (lines 357-365) sections were revised to enhance clarity for readers.

Reviewer 3 Report

Comments and Suggestions for Authors
  1. Supplement the calculated average values described in Lines 201 to 206 into Table 1 to make data comparison more intuitive. Additionally, supplement the reference sources for the data of each group
  2. The differences in the expression of relevant enzymes among different lighting groups shown in Figure 2 are not very clear. It is recommended to label specific details of lighting conditions and other relevant information.
  3. When citing some experiments, only the light intensity is described, while details such as the broilers' age, sex, and breed are not provided. However, these factors can also have an impact on the results, so further supplementation of this information is necessary.
  4. In the article, some mechanistic studies are mentioned, such as how melatonin levels improve broiler welfare and metabolic function, as well as the mechanisms of action of factors affecting bone development and muscle repair. However, these explanations are relatively general. Further supplementation of research evidence or hypotheses regarding molecular mechanisms is recommended to enhance the scientific rigor and depth of the article.

Author Response

Dear Reviewer:

We appreciate your helpful and detailed feedback and comments, and hope that the explanation below has thoroughly addressed all your comments. In this response, we discuss each of your comments individually, along with corresponding responses.

In the revised manuscript, we used the latest version of the manuscript (manuscript v3.docx).  Two tables were converted into editable MS Word tables, rather than images. Two additional references were added, and one reference was changed in position. All references were confirmed. Yellow-marked words or paragraphs were the ones that were changed in the revised manuscript.

# Reviewer 3

Comment 1: Supplement the calculated average values described in Lines 201 to 206 into Table 1 to make data comparison more intuitive. Additionally, supplement the reference sources for the data of each group.

Response 1: Thank you for pointing this out. The final days of each experiment (trial) were varied across trials due to the commercial industry's schedules. Therefore, we should consider the treatment effects on daily body weight gain (DWG) and feed conversion ratio (FCR) by percentage changes instead of DWG (g) and FCR by itself, because of the different final days. In the revised manuscript, values of DWG and FCR are compared among treatments by percentage differences, and three references were provided, lines 215-220.

Comment 2: The differences in the expression of relevant enzymes among different lighting groups shown in Figure 2 are not very clear. It is recommended to label specific details of lighting conditions and other relevant information.

Response 2: Appreciate this comment. In the revised manuscript, Figure 2 has improved in quality for readers between lines 275 and 276. Specific details of lighting conditions and relevant information are added in lines 277-280 of the Figure legend.

Comment 3: When citing some experiments, only the light intensity is described, while details such as the broilers' age, sex, and breed are not provided. However, these factors can also have an impact on the results, so further supplementation of this information is necessary.

Response 3: I agree that the broiler’s age, sex, and breed can also have an impact on the results. We observed a slight difference in results between Cobb 700 and Ross 308. However, variable-lighting and enrichment hut studies in commercial broiler houses were mainly investigated using Cobb 700 mixed-sex birds. This information was added to line 183 of the revised manuscript.

Comment 4: In the article, some mechanistic studies are mentioned, such as how melatonin levels improve broiler welfare and metabolic function, as well as the mechanisms of action of factors affecting bone development and muscle repair. However, these explanations are relatively general. Further supplementation of research evidence or hypotheses regarding molecular mechanisms is recommended to enhance the scientific rigor and depth of the article.

Response 4: Authors agree on the generalization of the explanation regarding the effects of melatonin. Currently, we lack additional research evidence on the molecular mechanism of melatonin as an antioxidant during daytime resting /sleeping in poultry studies. Studies indicate melatonin's impact on skeletal muscle and exercise in rodents (Antic et al., 2025; Stacchiotti et al., 2020). In this article, we speculated on the possible mechanism of the beneficial effects of daytime resting/sleeping in the darker areas based on the mammalian studies to explain the results of the research trials in the commercial broiler houses. Nonetheless, authors think that sharing present information would be beneficial to the public readers and poultry scientists. Surely, there is a need for further research on the mechanism of melatonin on broiler welfare and production.

Round 2

Reviewer 2 Report

Comments and Suggestions for Authors

Thank you to the authors for making the changes and also for clarifying my comments. I am used to classic review formats, but I understand your point of view and believe that this article will significantly contribute to enriching practical knowledge in the field of broiler chicken welfare.